# Autophagy and Respiratory Viruses: Mechanisms, Viral Exploitation, and Therapeutic Insights

**DOI:** 10.3390/cells14060418

**Published:** 2025-03-12

**Authors:** Farnaz Aligolighasemabadi, Estera Bakinowska, Kajetan Kiełbowski, Mohammadamin Sadeghdoust, Kevin M. Coombs, Parvaneh Mehrbod, Saeid Ghavami

**Affiliations:** 1Division of BioMedical Sciences, Faculty of Medicine, Health Sciences Centre, Memorial University of Newfoundland, 300 Prince Phillip Dr., St. John’s, NL A1B 3V6, Canada; f.aligoli@outlook.com (F.A.); msadeghdoust@mun.ca (M.S.); 2Department of Human Anatomy and Cell Science, University of Manitoba College of Medicine, Winnipeg, MB R3E 3P5, Canada; esterabakinowska@gmail.com (E.B.); kajetan.kielbowski@onet.pl (K.K.); 3Department of Physiology, Pomeranian Medical University in Szczecin, 70-111 Szczecin, Poland; 4Department of Medical Microbiology and Infectious Diseases, University of Manitoba, Winnipeg, MB R3E 0J9, Canada; kevin.coombs@umanitoba.ca; 5Influenza and Respiratory Viruses Department, Pasteur Institute of Iran, Tehran 1316943551, Iran; mehrbode@yahoo.com; 6Paul Albrechtsen Research Institute, CancerCare Manitoba, University of Manitoba, Winnipeg, MB R3E 0V9, Canada; 7Akademia Śląska, Ul Rolna 43, 40-555 Katowice, Poland; 8Children Hospital Research Institute of Manitoba, University of Manitoba, Winnipeg, MB R3E 3P4, Canada

**Keywords:** adenovirus, autophagy, coronavirus, HPIV, influenza virus, respiratory syncytial virus (rsv), respiratory viruses

## Abstract

Respiratory viruses, such as influenza virus, rhinovirus, coronavirus, and respiratory syncytial virus (RSV), continue to impose a heavy global health burden. Despite existing vaccination programs, these infections remain leading causes of morbidity and mortality, especially among vulnerable populations like children, older adults, and immunocompromised individuals. However, the current therapeutic options for respiratory viral infections are often limited to supportive care, underscoring the need for novel treatment strategies. Autophagy, particularly macroautophagy, has emerged as a fundamental cellular process in the host response to respiratory viral infections. This process not only supports cellular homeostasis by degrading damaged organelles and pathogens but also enables xenophagy, which selectively targets viral particles for degradation and enhances cellular defense. However, viruses have evolved mechanisms to manipulate the autophagy pathways, using them to evade immune detection and promote viral replication. This review examines the dual role of autophagy in viral manipulation and host defense, focusing on the complex interplay between respiratory viruses and autophagy-related pathways. By elucidating these mechanisms, we aim to highlight the therapeutic potential of targeting autophagy to enhance antiviral responses, offering promising directions for the development of effective treatments against respiratory viral infections.

## 1. Introduction

Respiratory viruses present major global health challenges due to their high transmission rates and potential for severe diseases [1,2]. Despite vaccination programs, viral infections of the lower respiratory tract rank among the leading causes of death, alongside conditions like cancer, stroke, and diabetes [3,4]. Therefore, respiratory system-related viral infections are regarded as a significant health burden. Severe illness caused by these viruses can affect children, healthy adults, older adults, and immunocompromised individuals [5]. In addition, viral infections are responsible for up to 30% of community-acquired pneumonia cases in adults [6,7,8].

The identified viral causes in adults include influenza viruses (8%), rhinoviruses (6%), coronaviruses (3%), and respiratory syncytial virus (RSV) (2%); however, in pediatric patients with lower respiratory tract infections, RSV, influenza viruses, parainfluenza viruses, and metapneumovirus have been strongly linked to these illnesses [7,9]. A list of the most important respiratory viruses and their autophagy-related mechanisms is mentioned in Table 1. Even though respiratory viruses typically produce only mild cold and cold-like symptoms in healthy individuals, they still cause substantial productivity losses [10]. Viruses like influenza, which is known for its seasonal outbreaks, can lead to widespread illnesses and even death, especially among vulnerable populations like older adults and infants [11]. Influenza remains a significant cause of respiratory illness globally, causing an estimated 3–5 million cases of severe illness and 290,000–650,000 deaths annually worldwide [12]. RSV is the leading cause of acute lower respiratory tract infections in children under five and older adults, leading to over 33 million cases globally, with approximately 120,000 deaths annually [13]. It places a significant annual burden on the healthcare system [14]. COVID-19 has resulted in millions of infections and hundreds of thousands of deaths worldwide [15]. The recent COVID-19 pandemic, caused by severe acute respiratory syndrome coronavirus 2 (SARS-CoV-2), has reminded us of the profound global impact of respiratory viruses, showing how they can disrupt societies and economies on an unprecedented scale [16]. Parainfluenza viruses are also major contributors to respiratory infections, particularly in immunocompromised and older adults [17]. Adenoviruses can cause severe respiratory distress, especially in military populations and young children [18].

Although several viral families can potentially infect the respiratory tract, the treatment options are often restricted to supportive care. Recent advancements have enhanced our comprehension of the molecular mechanisms and cellular processes involved in respiratory viral infections and the host response [29].

Autophagy is a cellular process. Viruses are obligate parasites with no biosynthetic capacity when they are outside the host cells and with very simple, relatively tiny genomes. Therefore, without a host cell in which to replicate, viruses do not themselves experience autophagy. However, many of them strongly induce and/or inhibit cellular autophagy during the infection/replication process as detailed below. A critical focus has been on autophagy, which is crucial for understanding the interactions between respiratory viruses and host cells [30,31,32]. Investigating the effects of autophagy on viral defense mechanisms, antiviral responses, and how viruses exploit autophagy-related pathways could provide valuable insights for the development of effective therapeutic strategies [32,33,34].

There are several types of autophagy, each with a distinct mechanism and function: microautophagy, macroautophagy, and chaperone-mediated autophagy (CMA) [35,36,37,38]. A schematic illustration of the autophagy pathways is shown in Figure 1.

Macroautophagy, commonly referred to as autophagy, is the most extensively studied form of autophagy [35,39,40]. This process involves several steps including initiation, phagophore expansion, autophagosome and lysosome fusion, and developing an autolysosome for cargo proteolytic destruction [21,36,41]. It is an essential cellular process that maintains homeostasis by breaking down and recycling damaged organelles, misfolded proteins, and other cellular debris [42,43]. It is vital for cellular quality control, energy balance, and stress response and plays a role in various physiological processes, such as development, immune responses, and aging [43,44,45]. Furthermore, autophagy can be selective under some conditions based on the specific targets, such as aggrephagy (protein aggregates), mitophagy (mitochondria), ciliophagy (cilia), and xenophagy (invasive microbes) [46,47,48,49].

In the xenophagy process, foreign molecules, rather than internal cellular components, are sequestered and degraded within autophagosomes [50].

On the other hand, dysregulation of autophagy is linked to numerous diseases including cancer, neurodegenerative disorders, and infectious diseases. Understanding the mechanisms and functions of autophagy offers valuable insight into its potential as a therapeutic target [21,51].

Autophagy is regulated by several key signaling pathways, including AMP-activated protein kinase (AMPK), mTOR, and ULK1. The mTOR pathway inhibits autophagy under nutrient-rich conditions by suppressing the ULK1 complex, while AMPK activates autophagy in response to cellular stress [52].

Recently, the dual roles of autophagy in unique virus infections have garnered wider recognition. Nevertheless, some viruses may hijack and exploit the autophagy machinery to favor their replication [53,54,55]. There are conflicting results that suggest viruses might induce or inhibit autophagy depending on the specific stage of infection. For example, RSV has been shown to inhibit autophagosome maturation by targeting Beclin1 [23], while coronaviruses enhance autophagy initiation but block the later stages to avoid degradation of viral components [56]. These mechanisms illustrate the complex interplay between autophagy regulation and viral pathogenesis.

This review highlights the dual roles of autophagy in the most important respiratory viral infections and host defense, pointing to the complex interplay between autophagy-related pathways and respiratory viruses’ life cycles. Then, we focus on the therapeutic potential of targeting autophagy, offering approaches to develop effective treatments against respiratory viral infections.

## 2. Influenza Virus and Autophagy

The *Orthomyxoviridae* family includes four subtypes of influenza viruses (A–D), three of which (A, B, and C) can infect humans. Influenza A and B are the primary causes of seasonal flu. Their RNA genomes encode at least 12 proteins, including the surface glycoproteins hemagglutinin (H) and neuraminidase (N). Influenza A is further classified into subtypes based on its H and N variants, such as H1N1 and H3N2, which currently account for the majority of seasonal infections [57].

Influenza remains a significant global health burden, leading to millions of infections annually and imposing high hospitalization costs, particularly in severe cases involving complications like myocarditis [58,59,60,61,62]. Over time, research has revealed that influenza viruses can evade the human immune system through various mechanisms, including manipulating autophagy. Autophagy plays a complex role in viral infections. While it serves as a defense mechanism, viruses have evolved to exploit it for their benefit. In the case of influenza, the virus enhances autophagosome accumulation, promoting its own replication [63,64]. However, stimulating lysosome–autophagosome fusion counteracts this effect, highlighting a potential therapeutic strategy for controlling viral infections [65]. In addition to the findings highlighted in previous studies regarding the interaction of influenza virus and autophagy [3], several other mechanisms were found to contribute to enhance replication of the virus depending on the subtype. In the H5N1 variant, the M2 and NP proteins play significant roles in controlling autophagy and viral replication. The viral NP interacts with microtubule-associated protein 1 light chain 3 beta (LC3) protein and increases its accumulation, while the viral M2 protein stimulates the budding of the virus [63,66].

LC3 is a central marker of autophagy. Upon autophagy induction, LC3-I undergoes lipidation through conjugation with phosphatidylethanolamine, forming LC3-II, which subsequently associates with autophagosomal membranes [67]. This LC3-I to LC3-II conversion is widely employed as a method to assess autophagic flux [68]. LC3-II facilitates the elongation and closure of the autophagosome and plays a role in cargo selection by interacting with autophagy receptors such as p62/SQSTM1 [69].

Upregulation of HSP90AA1 and the involvement of the AKT/mTOR signaling pathway are required for autophagosome accumulation [63]. Moreover, HSP90AA1 is one of the binding proteins for influenza [70]; thus, the use of HSP90AA1 targeting antibody could reduce viral entry, prevent the induction of autophagy, and suppress viral replication. Additionally, HSP90AA1 stabilizes the PCBP1-AS1-encoded small protein (PESP), a recently discovered protein, overexpression of which regulates autophagy and enhances influenza replication [71]. Regarding influenza A (H1N1) pdm09 subtype, the non-structural protein 1 (NS1) antagonizes leucine-rich pentatricopeptide repeat-containing protein (LRPPRC), which is crucial in the viral replication process [64]. LRPPRC negatively regulates autophagy by interacting with BECN1 (Beclin1), a key protein involved in autophagy initiation. In a normal physiological state, LRPPRC forms a complex with BECN1 that inhibits the autophagic pathway [72,73]. The NS1 protein disrupts the LRPPRC complex by binding to it, which counteracts its repressive function. This disruption enhances the interaction between BECN1 and PIK3C3, a key enzyme in autophagosome formation necessary for cellular degradation [64]. As a result, the antagonism of LRPPRC by NS1 facilitates the activation of autophagy through increased recruitment of BECN1 by PK3C3 [64]. Apart from stimulating autophagy, infection with the H9N2 variant induces oxidative stress, which in turn enhances necroptosis due to bacterial co- or secondary infection [74,75]. Thus, apart from reducing viral replication, suppression of autophagy could suppress the severity of bacterial co-infection.

The virus alters autophagy to suppress the host’s immune response. Several recent studies demonstrated that influenza interacts with mitochondrial antiviral signaling protein (MAVS) to regulate interferon (IFN) responses. IFN alerts uninfected cells about the presence of the virus and induces antiviral mechanisms. The inflammatory responses stimulated by IFN enhance “flu-like” symptoms [76]. Viral proteins can stimulate MAVS degradation and enhance interaction with mitochondrial LC3B, stimulating mitophagy [77,78]. Enhanced MAVS degradation is associated with increased viral replication [79]. Studies demonstrated that mitophagy is enhanced due to the activity of the PB1-F2 [77] and NP [78] proteins. The virus can also exploit autophagy to enhance inflammatory responses. Researchers demonstrated that stimulation of autophagy and secretion of exosomes increase the presence of M1 pro-inflammatory macrophages [80], which were previously associated with a severe course of infection [81]. By inhibiting autophagosome–lysosome fusion, infection with influenza A virus (IAV) might prevent efficient loading of viral antigens into the MHC class II molecules. Autophagosome protein composition might be affected by the virus, which impairs the processing of the antigen [82]. Moreover, by interfering with antigen presentation, IAV might impair differentiation and activation of the T-cells [83,84].

Importantly, the host cells can use autophagy to counteract the harmful effects of the virus. In the case of the avian influenza H7N9 subtype, Liu et al. found that p62 can contribute to the formation of viral RNA aggregates in the cytoplasm, which then are diverted to degradation [85].

As autophagy seems to play a dual role in viral infections, researchers have been examining the use of various agents to modulate autophagy to suppress the infection. By modulating autophagy, gallic acid [86], vitamin D3 [87], baicalin [88], tanreqing [65], and the Huanglian-Ganjiang combination [89] could suppress influenza virus infection. The modulation of autophagy presents a promising therapeutic strategy for combating influenza infection, as outlined in Table 2. Gallic acid has been shown to decrease viral load by preventing excessive autophagosome accumulation, thereby reducing viral exploitation of the autophagic process [86]. Similarly, vitamin D3 enhances autophagic flux by promoting autophagosome–lysosome fusion, facilitating viral degradation, and ultimately decreasing viral replication [87]. In addition, baicalin contributes to improved macrophage viability by downregulating autophagy markers, suggesting a role in maintaining balanced autophagic response [88]. Likewise, tanreqing has been found to inhibit viral replication by enhancing autophagosome–lysosome fusion, further supporting the notion that facilitating autophagic degradation is a viable antiviral approach [65]. These findings underscore the therapeutic potential of autophagy modulation in influenza treatment.

Targeting specific autophagy pathways holds great therapeutic potential. By modulating autophagy, the ability of the virus to exploit autophagy for its replication could be suppressed. Viral M2 was shown to induce the formation of autophagosomes. By targeting M2 with a compound that blocks its interaction with the proteins associated with autophagy, virus replication might be decreased. Rapamycin, which is an mTOR inhibitor, induces autophagy. However, as viruses may sometimes benefit from enhanced autophagy, using rapamycin as a therapeutic option should be considered with caution [90].

Interestingly, several natural compounds hold great promise in treating viruses. Astragaloside IV impairs autophagosome accumulation targeted by the virus and increases autophagic flux for viral particle degradation [29]. Additionally, virus-induced autophagy and mRNA synthesis were suppressed by aloe vera ethanol extract [91]. Figure 2 shows a schematic illustration of influenza virus interaction with autophagy.

Apart from autophagy, influenza virus affects other processes. For instance, IAV affects apoptosis through the viral NP protein that triggers intrinsic apoptosis via the mitochondrial pathway. Other viral proteins M1, M2, NS1, and PB1-F2 modulate cell death pathways [92]. Furthermore, IAV activates UPR in several mechanisms. For instance, the NS1 protein reduces host proteins resulting in ER stress [93]. Moreover, apoptosis promoted by enhanced autophagy might contribute to organ dysfunction and tissue damage [87,94].

Several aspects of interactions between influenza and autophagy are not fully elucidated. While most of the studies were carried out on A549 cells, experiments implementing human airway epithelial cells or immune cells should be performed. Furthermore, regulation of autophagy by proteins other than M1, M2, and NS1 remain not fully understood [70,90]. Moreover, the interactions between autophagy and viral antigen presentation and adaptive immune responses require further investigation [26].

To summarize, influenza and host mediate autophagy in the following mechanisms: (i) influenza enhances autophagosome accumulation to increase its replication; (ii) the virus promotes pro-inflammatory immune responses, which can be associated with more severe course of infection; (iii) influenza stimulates mitophagy to evade the innate immune response by suppressing interferon responses; and (iv) autophagy proteins can form vRNA aggregates to counteract harmful effects of the virus.

Based on the current evidence, agents that enhance autophagosome–lysosome fusion and those that suppress mitophagy might be associated with improved antiviral responses.

## 3. Respiratory Syncytial Virus and Autophagy

Respiratory syncytial virus (RSV) is a single-stranded RNA virus belonging to Pneumoviridae family, genus *Pneumovirus*. The size of the virus ranges between 150 and 300 nm and its genes are arranged in order from 3′ to 5′ in the following way: NS1-NS2-N-P-M-SH-G-F-M2-L [95,96]. RSV is responsible for lower respiratory tract infections (LRTI) such as bronchiolitis and pneumonia. It affects people of all ages and is a significant burden especially for young children, older adults with comorbidities, and immunocompromised individuals. Each year, LRTI causes 3–6 million hospitalizations and over 118,000 deaths among children younger than 5 years. There is no specialized treatment except for supportive care [13]. Studies suggest that RSV infection during childhood can contribute to the development of asthma and recurrent wheezing later in life [97,98].

Cellular entry of the virus is mediated by insulin-like growth factor-1 receptor (IGF1R) and nucleolin (NCL). The binding of RSV with former receptors stimulates NCL trafficking towards the cell membrane, thus allowing the virus to bind to NCL and enter the cell. Mechanistically, interaction between RSV and IGF1R stimulates the activity of protein kinase C zeta (PKCζ), which was found to mediate the effect on NCL [99]. Further experiments confirmed that throughout the infection, RSV downregulates IGF1R expression on human airway epithelial cells. Nevertheless, this observation is thought to promote viral infection as it aids the release of viruses [100]. Recently, Kieser et al. described the importance of actin remodeling in mediating RSV cell entry. Actin rearrangements induced by the virus promote macropinocytosis and mediate NCL trafficking to be able to bind to RSV. Researchers demonstrated that phosphoinositide 3 kinase (PI3K), one of the downstream signaling element of IGF1R, regulates the observed effects [100]. Therefore, current evidence demonstrates complex involvement of IGF1R and its signaling pathways in the process of RSV cellular entry. These findings offer opportunities for targeting RSV–IGF1R to prevent infection. CL-A3-7 is a small molecule that suppresses RSV infection in experiments in vitro. Mechanistically, the molecule was found to suppress the viral F protein and IGF1R, thus limiting viral entry and infection [101]. As the virus interacts with cellular receptors through the F protein, targeting binding with IGF1R could modulate the activity of preF antigen vaccines. The vaccine works by stimulating the production of antibodies targeted at perfusion conformation of the F protein. Recent clinical trials showed that the vaccine is well tolerated and efficacious in stimulating the immune system against RSV [102,103]. Perhaps, the combination of preF vaccine with an inhibitor of IGF1R could enhance the antiviral response and synergistically suppress the RSV ability to enter the cells. Another method to target RSV host invasion is through direct targeting of F protein. Ziresovir (AK0529, RO-0529) is a selective RSV F protein inhibitor. Ziresovir exerts its antiviral effect by targeting the RSV F protein. It effectively blocks the virus ability to penetrate human cells, thereby preventing the initiation and spread of RSV infection [104]. Ziresovir demonstrated a favorable safety and tolerability profile in hospitalized infants with RSV infection. The treatment was well tolerated, with no treatment discontinuations or deaths [105]. It is an open question whether F protein inhibitors could be combined with suppression of IGF1R activity.

Each of the viral proteins plays a crucial role during attachment to the host cells, replication, and infectivity. The study by Han et al., demonstrated that NS1 protein of RSV induces the autophagy pathway through inhibition of the mTOR-S6KP70 signaling pathway. The production of IFN-α and inflammatory cytokines as well as the activation of apoptosis was inhibited by the enhanced process of autophagy providing a beneficial environment for the replication of the virus [106]. In line with this finding, a study by Liu et al. suggested that inhibition of the mTOR pathway increases the amount of autophagosomes in bronchial epithelial cells [107]. By contrast, Azman and colleagues indicated that pharmacological inhibition of autophagy did not alleviate inflammation in RSV-infected human epithelial cells, which contradicts the theory proposed in other studies using mouse models [108]. Moreover, the inhibition of NS1 has been proven to be a considerable protection from inflammation caused by RSV infection. The viral NS1 protein holds great promise for therapeutic applications in viral infections [109]. The non-structural protein 2 (NS2) mediates autophagy induced by RSV through the stabilization of Beclin1 by escaping proteasome degradation. The NS2 reduces the function of interferon-stimulated gene 15 (ISG15) via Beclin1 ISGylation. This interaction forms active Beclin1 by its hypo-ISGylation for the successful induction of autophagy [23].

Qingfei (QF) oral liquid, a traditional Chinese medicine, is used to treat asthma and pneumonia. It was discovered that QF inhibits the formation of autophagosomes in asthmatic mice infected with RSV, which alleviated inflammation [110]. A study by Lin and collaborators reported that QF reduced inflammation caused by viral-associated autophagy via reduction of F and G protein expression [111].

Rapamycin, also known as sirolimus, is an immunosuppressant and autophagy activator [112]. Studies have demonstrated that it can suppress overactive inflammatory reactions during viral infections, such as those caused by influenza viruses [113]. The recent study by Huckestein et al. reported that rapamycin treatment of influenza-infected mice resulted in decreased pulmonary inflammation and lowered the prevalence of exudate macrophages and B and T lymphocytes. It was suggested that rapamycin treatment may reduce activation of mTORC1 after influenza infection [114]. Moreover, rapamycin has shown potential in suppressing cytokine storms in COVID-19 patients by reducing the secretion of various cytokines, such as IL-2, IL-6, and IL-10. Many studies have demonstrated rapamycin inhibitory effects on IL-10 mRNA and protein levels, as well as its ability to suppress STAT-3, a key regulator of IL-10, and other signaling pathways. This suggests that rapamycin could be a valuable treatment in controlling the excessive inflammatory responses associated with severe COVID-19 cases [115]. However, much less is known regarding the potential activity of rapamycin in RSV. Rapamycin is an agent with pleiotropic effects and can influence a variety of immune cells. In murine bone marrow-derived dendritic cells, rapamycin enhanced viral replication and reduced their ability to differentiate CD8+ T cells [116].

Chloroquine (CQ), which is known for its anti-malarial properties, has demonstrated properties of blocking autophagy. Autophagy flux is blocked by impaired fusion of autophagosome and lysosome caused by CQ [117]. A study by Wang et al. reported that CQ successfully inhibited the novel coronavirus in an in vitro experiment. CQ increases pH in endosomes that prevents fusion of virus and cells, which allows it to block the potential infection [118].

Metformin is a relatively old therapeutic given to patients affected by type-2 diabetes mellitus, which is also known for its pleiotropic effects. For instance, it modulates the AMPK-mTOR pathway [119]. In the context of SARS-CoV-2, metformin may exhibit antiviral properties by preventing viral entry into host cells. This occurs through the AMPK-mediated phosphorylation of the angiotensin-converting enzyme 2 (ACE2) receptor at serine 680 (S680). This phosphorylation induces changes in the ACE2 receptor, potentially reducing the binding affinity of the virus [120]. Moreover, pH regulation in endosomes is important for viral infection. Vacuolar ATPase (V-ATPase) and endosomal Na+/H+ exchangers (eNHE) are involved in the process of pH regulation in endosomes. It was reported that metformin may directly impact V-ATPase and/or eNHE, resulting in increased pH which may suppress viral infection [121,122]. In the study by Parthasarathy et al., the authors indicated that metformin significantly decreased the growth of SARS-CoV-2 in cell culture models. An increase in AMPK phosphorylation was observed during the progression of the viral infection. It was reported that metformin suppressed SARS-CoV-2 infectiousness up to 99% in naïve and infected cells [123]. The recent clinical trial by Carolyn and colleagues confirmed that metformin lowered viral load 4.4-fold when compared to a placebo group. On the 10th day of the study, a statistically significant difference was observed between the metformin and placebo groups. The undetectable rate in the metformin group was 14.3%, compared to 22.6% in the placebo group [104]. Therefore, several drugs that regulate autophagy and viral infections are known. Nevertheless, more studies are required to understand their efficacy against RSV infection.

A recent study by Chen et al. revealed that RSV enhances accumulation of cholesterol in lysosomes by inhibiting their transport to the endoplasmic reticulum (ER) through reducing lysosomal acid lipase activity. The elevated levels of cholesterol impair VAP-A and ORP1L binding, and dynein–dynactin recruitment enables the formation of autolysosome and autophagosome transportation. Inhibition of lipase activity and lysosomes with a great amount of cholesterol impairs autophagy by blocking autophagosome degradation, leading to the accumulation of RSV fusion protein to ensure effective viral replication. Furthermore, knockout of the low-density lipoprotein receptor (LDLR) inhibited both in vitro and in vivo RSV infection by mediating lysosomal cholesterol metabolism and autophagy. The knowledge about this regulation may be used in the development of anti-RSV drugs by targeting LDLR [124].

Figure 3 shows a schematic illustration of RSV interaction with autophagy. Apart from this regulation, targeting other crucial autophagy factors serves as a promising therapy strategy. For instance, AMPK activation may induce autophagy by inhibiting the mTOR pathway. Procyanidin A1 and Trifolirhizin are AMPK inducers that are currently under preclinical trials, serving as promising therapeutic options [125,126,127].

There was an urge to develop RSV vaccines to prevent the spread of the virus and serious complications related to the infection. In 2023, two protein subunit vaccines were licensed for severe RSV for patients over 60 years old. The vaccines induce immune responses by targeting RSV preF protein [102,128].

According to the study investigating the effect of RSV vaccination in older and younger populations, CD8+T cells presented reduced levels of autophagy in older adults. It was suggested that the autophagy process is involved in the efficacy of the vaccine in older adults [129]. Moreover, antigen presentation induced by autophagy enhanced the production of cytokines and T-cell activation [130].

## 4. Coronaviruses and Autophagy

Following the emergence of SARS-CoV in 2002, Middle East respiratory syndrome coronavirus (MERS-CoV) in 2012, and SARS-CoV-2 in 2019, the world has witnessed the significant impact of coronaviruses on global public health [32,54,131]. Coronaviruses, including SARS-CoV, MERS-CoV, and SARS-CoV-2, are positive-sense, single-stranded RNA viruses that possess the largest RNA genomes among viruses infecting mammals [132]. Coronaviruses infect a variety of birds and mammals, including humans, leading to diseases that primarily impact the respiratory, intestinal, and nervous systems [133,134,135]. Coronaviruses like SARS-CoV-2 bind to the angiotensin-converting enzyme 2 (ACE2) receptor via their spike protein to enter the host cells, where they release their RNA and hijack the host machinery to produce viral proteins and replicate [136,137]. The new virions are assembled in the ER–Golgi compartment, then released and spread the infection [137]. The immune response, including cytokine release, can lead to severe inflammation, tissue damage, and complications such as acute respiratory distress syndrome (ARDS) and multi-organ failure [138].

Given the significant morbidity and mortality caused by these viruses, understanding the underlying mechanisms of their pathogenesis has become a critical area of research.

Among the various aspects studied, recent investigations have particularly highlighted the complex interplay between autophagy and coronavirus infections, especially in the cases of SARS-CoV-2 and MERS-CoV.

SARS-CoV-2 has been observed to interact with autophagy at the beginning of its life cycle. It has been shown that ACE2, a primary receptor for SARS-CoV-2, plays a crucial role in facilitating viral entry into the host cells, while also acting as a cellular receptor that suppresses cell apoptosis and inhibits autophagy in the lungs [139,140]. Furthermore, SARS-CoV-2 lowers the levels of essential proteins needed for the early phases of autophagy, including BECN1, Class III Phosphatidylinositol 3-Kinase (VPS34), and Autophagy-Related Gene 14 (ATG14) [141]. This disruption hinders the development of the phagophore [141]. This suggests that SARS-CoV-2 may engage with autophagic machinery upon attachment, potentially using the autophagy pathway to assist in its entry and replication. Figure 4 shows a schematic illustration of SARS-CoV-2 interaction with autophagy.

Interestingly, it has been reported that atorvastatin may induce autophagy through multiple pathways [142]. Notably, atorvastatin upregulates the expression of autophagy-related markers such as Beclin1, p53, and LC3-II at both the mRNA and protein level. Additionally, it activates the AMPK/mTOR pathway, a well-established regulator of autophagy [143,144,145].

It has been observed that various SARS-CoV-2 proteins can trigger autophagy through distinct mechanisms in vitro. This finding was supported by Qu et al., who further identified that the viral protein ORF3a plays a crucial role in autophagosome formation, facilitating the virus replication cycle [146]. Similarly, Hayn et al. observed that ORF3a inhibits the fusion of autophagosomes with lysosomes, leading to the accumulation of autophagosomes, which aids viral survival [147]. In a separate study, it was demonstrated that ORF3a achieves this inhibition by blocking the assembly of the STX17-SNAP29-VAMP8 SNARE complex, a critical step in autophagosome–lysosome fusion [148]. ORF7a also disrupts the initiation phase of autophagy by decreasing lysosomal acidity, which hinders the cell’s capacity to break down components through autophagy [149].

Sun et al. reported that SARS-CoV-2 non-structural protein 6 (NSP6) restricts autophagosome expansion [150]. By limiting the size of autophagosomes, SARS-CoV-2 effectively reduces the presentation of viral antigens, thereby diminishing the activation of the adaptive immune response [151]. NSP6 overexpression in lung epithelial cells triggers inflammasome activation, caspase-1-dependent pyroptosis, and autophagic flux inhibition by disrupting lysosome acidification through its interaction with ATP6AP1. A variant of NSP6 (L37F), associated with asymptomatic COVID-19, shows reduced binding to ATP6AP1 and diminished ability to impair autophagy, highlighting potential therapeutic approaches.

Shang et al. observed that SARS-CoV-2 infection triggers the activation of ULK-1–Atg13 and VPS34–VPS15–BECN1 complexes, which facilitates the formation of autophagosomes [152], a finding that contrasts with that of Kumar et al., who noted the viral protein NSP6 impedes the initiation of autophagy by disrupting the assembly of pre-autophagosomal structures [153]. In this context, Zhang et al. demonstrated that coronavirus non-structural protein 15 (Nsp15) inhibits the host’s innate immune response by preventing the nuclear translocation of phosphorylated IRF3 [64]. Nsp15 achieves this by binding to the nuclear import adaptor karyopherin α1 (KPNA1) and promoting its degradation through autophagy to block the induction of type I interferon response [64].

Another study by Feng et al. found that MERS-CoV uses a distinct approach by disrupting autophagic flux [154]. The virus Nsp1 downregulates the mRNA of lysosome-related genes, resulting in decreased lysosomal biogenesis and acidification. This disruption of autophagic flux aids in viral survival and facilitates replication [155]. In addition to ORF3a, NSP6 that is crucial for preventing autophagosome–lysosome fusion also disrupts interactions between the SNARE complex proteins STX17, VAMP8, and SNAP29, which are necessary for this fusion. As a result, autophagosomes accumulate, hindering the final stages of the autophagic process [155]. Overall, increased autophagy has the potential to decrease MERS-CoV replication, suggesting that autophagy could serve as a novel therapeutic target for managing MERS-CoV infection.

Further studies on respiratory infections caused by MERS-CoV showed that the virus induces AKT1 activation through phosphorylation, which in turn activates S-phase kinase-associated protein 2 (SKP2) [156]. SKP2 is part of the SCF ubiquitin–ligase complex and plays a role in regulating autophagy by targeting specific proteins for degradation [157]. SKP2 can influence the levels of autophagy-related proteins, such as p27Kip1, thereby modulating autophagic activity and affecting cell cycle progression. Dysregulation of SKP2 has been linked to various diseases, including cancer, where it may contribute to the altered autophagic processes that help cancer cells survive under stress [146]. Understanding the relationship between SKP2 and autophagy may offer insights into potential therapeutic strategies for diseases with autophagic dysregulation. Activation of SKP2 can also lead to the degradation of BECN1, inhibiting the fusion of autophagosomes with lysosomes [156]. This disruption may help protect viral replication complexes situated on the cellular double-membrane structures.

Coronaviruses such as SARS-CoV-2 and MERS-CoV engage in a complex relationship with autophagy, employing various strategies to exploit or inhibit this cellular process to support their survival and replication. Therefore, targeting autophagy emerges as a promising therapeutic approach, with potential strategies that could effectively combat coronavirus infections and mitigate their significant public health impact. Further research is vital for understanding the intricate interactions between autophagy and coronaviruses, which could enhance our knowledge of viral pathogenesis and lead to the development of novel interventions.

Three categories of autophagy modulators can inhibit viral replication [158]. The first category consists of drugs with lysosomotropic properties. Chloroquine and hydroxychloroquine, two widely studied drugs in this category, increase lysosomal pH and inhibit cathepsin activity, thereby blocking viral–endosomal fusion and reducing viral replication. Additionally, they prevent coronavirus infection by neutralizing the acidic pH of endosomes and lysosomes [159,160,161]. This disruption effectively blocks the fusion of the virus with the host cell membrane, halting the infection process at an early stage [162].

The second category comprises protease inhibitors, which can effectively prevent the proteolytic cleavage of the S protein, thereby inhibiting the virus’s ability to enter and infect host cells. Lopinavir and ritonavir, two well-known HIV protease inhibitors, have been repurposed for COVID-19 treatment due to their ability to suppress SARS-CoV-2 replication and reduce viral load [161,163,164].

The third category includes PI3K/mTOR regulators that, while regulating autophagy, can prevent coronavirus-mediated appropriation of the autophagic machinery. Rapamycin, an mTOR inhibitor, has been shown to reduce viral replication by suppressing mTORC1 activity, ultimately enhancing autophagy and limiting infection in MERS-CoV and SARS-CoV-2 models. In contrast, Nitazoxanide, an antiparasitic drug with antiviral properties, stimulates autophagy by blocking mTORC1 and inhibiting MERS-CoV and SARS-CoV-2 replication. Interestingly, Wortmannin suppresses autophagy and has been found to inhibit MERS-CoV infection effectively [114,158,165]. Table 3 summarizes various autophagy-related medications and their impacts on coronaviruses.

## 5. Human Parainfluenza Viruses and Autophagy

Human parainfluenza viruses (HPIVs) are enveloped, negative-sense, single-stranded RNA viruses that belong to the Paramyxoviridae family. There are four serotypes of the virus: HPIV1, HPIV2, HPIV3, and HPIV4 with the latter further subdivided into HPIV4a and HPIV4b. HPIV encodes six structural proteins, including fusion protein (F) and hemagglutinin-neuraminidase protein (H)-two glycoproteins, RNA polymerase (L), matrix protein (M), nucleocapsid protein (N), and phosphoprotein (P) [177]. Viral particles are pleomorphic and approximately 150–200 nm in size. Infection with HPIV has several clinical manifestations, which depend on the serotypes: HPIV1 and HPIV2—croup, HPIV3—bronchiolitis and pneumonia, HPIV4—bronchiolitis and pneumonia [17,178,179,180].

HPIV infections represent a significant burden for the global healthcare system. In a 12-year retrospective study, the rates of croup- and pneumonia-related costs for hospitalizing children under 5 years old were estimated at USD 58 million and USD 158 million, respectively [181].

These RNA viruses use the autophagy processes for their replication [182]. The HPIV3 phosphoprotein (P) is responsible for blocking degradation of the autophagosome. Localized in the external membrane of completed autophagosomes, Syntaxin17 (STX17) interacts with the SNAP29 protein belonging to the SNARE complex. SNAP29 interacts with vesicle-associated membrane protein 8 (VAMP8) in the lysosome membrane. P protein binds to SNAP29, which prevents its interaction with STX17, thus inhibiting fusion of autophagosomes with lysosomes. Insufficient autophagy leads to accumulation of autophagosomes, which elevates extracellular viral production. However, this process does not affect intracellular replication of the virus or production of viral proteins [25,183,184].

Different studies suggested that autophagy may enhance replication of the virus by inhibiting innate immunity or by stimulating the translation of viral proteins [185,186,187]. HPIV3 is recognized by the infected cells through retinoic acid-inducible gene I (RIG-I)-like receptors (RLRs), which play crucial roles during viral infection through the induction of type I interferons and other factors mediating immune responses. RIG-I has a caspase recruitment domain (CARD), which binds with MAVS. Interaction between the MAVS and CARD in RIG-I recruits inhibitors of NF-kB kinase (IKK) and TNF receptor-associated factor (TRAF), which stimulates production of IFN and pro-inflammatory cytokines, such as IL-18 and IL-1β [188,189]. The HPIV3 M protein, which is associated with regulation of replication and transcription of the virus, is involved in mitophagy [190,191]. During viral invasion, the M protein is translocated into the mitochondria through binding with TU translation elongation factor mitochondrial (TUFM). The process of mitophagy requires kinase PINK1 to label the impaired mitochondria that recruits E3-ubiquitin ligase Parkin, causing mitochondrial sequestration through autophagosome. However, the M protein acts as a mitophagy receptor, enhancing mitophagy independently from the Parkin–PINK1 pathway. The interaction between M and LC3 mediates the formation of autophagosomes and mitochondrial sequestration. According to the previously mentioned study, viral P protein prevents mitophagosomes from their fusion with lysosomes, resulting in incomplete mitophagy mediated by HPIV3. This process inhibits the RIG-I signaling pathway, thus suppressing type I IFN production. As a result, it prevents the expression of the IFN-stimulated genes responsible for blocking replication of the virus. Accumulation of mitophagosomes may serve as either a membrane or transportation depot for the virus [192,193]. Ferritinophagy is a type of autophagy responsible for the degradation of ferritin [194]. Nuclear receptor coactivator 4 (NCOA4) enables ferritin transportation to autophagosome vesicles. Fusion with lysosome degrades ferritin, leading to the release of iron ions into the cytoplasm. Their presence may be used for heme synthesis or other synthetic pathways. However, iron can also enhance the generation of reactive oxygen species (ROS) that stimulate apoptosis. By contrast, the process stimulates the functionality of the mitochondria in a state of iron deficiency [195,196,197].

The HPIV2 genome encodes the V protein that interferes with several host proteins such as STAT, TRAF6, and Caspase1 [198,199,200]. The HPIV2 V protein suppresses the interaction between NCOA4 and ferritin heavy chain (FTH1). This interaction prevents the degradation of vesicles by lysosomes and, subsequently, apoptosis. Additionally, it prevents HPIV2 degradation in host cells allowing the virus to grow and replicate effectively. Interestingly, it has been suggested that the growth of the virus is enhanced in an iron-rich environment [201,202]. An inhibited process of ferritinophagy leads to insufficient degradation of the virus as ROS are not generated by the excess of iron ions. It also affects the function of the mitochondria and oxygenation as decreased levels of iron would not be able to cover the need for heme production [203]. Figure 5 shows a schematic illustration of HPIV interaction with autophagy.

In conclusion, autophagy appears to be an important component for HPIV’s replication. Many studies have shown that virus proteins serve as valuable targets in antiviral therapy, particularly through inhibiting the autophagic pathway. Investigation of HPIV M and V proteins may serve as potential targets for vaccine and/or antiviral drug development.

## 6. Adenovirus and Autophagy

Adenoviruses (AdVs) are non-enveloped, double-stranded DNA viruses, which belong to the Adenoviridae family. The AdV size ranges from 70 to 100 nm. Over 50 serotypes and seven sub-families (A–G) have been identified [204,205]. These viruses often cause mild infections of the lower and upper respiratory tracts (C and E), keratoconjunctivitis (D), and infections of the gastrointestinal tract (F) [206,207]. The majority of cases are identified in children, where respiratory tract infections caused by AdVs account for 7-8% of pediatric infections due to the absence of humoral immunity [18]. Latent AdVs are found in many tissues such as renal parenchyma or lymph nodes [208,209]. Depending on the virus serotype and transmission mechanism, the estimated incubation time ranges between 2 and 14 days [205].

Attachment of the virus to host cells results in a rapid receptor-mediated endocytosis process that releases internal membrane lytic capsid protein VI (PVI). PVI damages the cellular membrane and allows the AdV to enter the cytosol and, consequently, the nucleus by microtubule transportation. Rupture of the membrane caused by AdV is recognized by galectins. Galectin-8 (Gal-8) and LC3 recruit impaired endosomes, which induces autophagic response.

A study by Wodrich et al. showed that increase in infectivity of the virus was correlated with the depletion of Gal8. Recruitment of Gal8 to the impaired membranes by viral infection enables autophagic response in infected cells by removing pathogens and damaged membrane [210,211]. GTPases regulate the function and identity of endosomes while Rab-proteins such as Rab7 and Rab5 are associated with the late and early endosomes, respectively [212]. Rab5 is a multifunctional protein that plays a significant role during endocytosis. It is responsible for regulating the fusion of early endosomes and the maturation of phagosomes [213]. Rab7 is associated with the late endocytic pathway compartments, such as lysosomes and endosomes. Rab7 enables the maturation of early endosomes into late endosomes and the formation of lysosomes and their fusion with late endosomes [214,215].

The AdV escapes autophagy if this concurs with Rab5–Rab7 exchange during the transition from early to late endosomes. Moreover, autophagy might inactivate the virus by the specific modulation of endosomes undergoing the above mentioned process [216].

Intriguingly, Rab5c overexpression enhances the replication of AdV and its inhibition blocks replication of the virus. It was speculated that promotion of AdV replication was the effect of the autophagy process rather than endocytosis. Rab5c interacts with Beclin1 and promotes LC3-II protein expression. This interaction induces complete autophagy, resulting in increased replication of the virus [217].

Montespan and colleagues reported that the AdV escapes autophagic degradation through the PPxY motif in PVI after endosomal lysis. The PPxY motif sequestrates Nedd4.2, which is a ubiquitin ligase, preventing autophagosome development and enhancing infectivity [218]. In the study by Zhang et al., the authors demonstrated that infection by human adenovirus B7 (HAdC-B7) activated autophagic flux. Fusion of lysosomes and autophagosomes induced by HAdV-B7 PVI suppressed the replication of the virus [219]. The host protein Bcl-2-associated athanogene 3 (BAG3) WW domain interacts with the PVI protein of the virus. Additionally, PVI of HAdV-B7 promotes the expression of BAG3 in infected cells [220]. Figure 6 shows a schematic illustration of AdV interaction with autophagy.

Interestingly, autophagy induction by rapamycin reduced HAdV-B7 production, while 3MA, which blocks the formation of autophagosomes, enhanced replication of the virus [219]. Zeng et al. indicated that Ad2 infectivity may be increased by autophagy. Autophagy upregulated the expression of the adenovirus early region 1A (E1A), a gene expressed early in the process of replication, which enables production of E1A proteins that permit replication of the virus in infected airway epithelial cells [216,221].

## 7. Potential Therapeutic Approaches Using Autophagy

Autophagy-modulating therapies are designed to either suppress or boost autophagic activity based on the therapeutic goal. For respiratory viruses, enhancing autophagy could aid in clearing viral particles and reducing inflammation, while its inhibition might prevent viruses from exploiting this pathway for replication [222,223]. This dual functionality makes autophagy a promising target for therapeutic interventions in respiratory viral infections [222,223].

The use of autophagy-modulating medications in the clinical setting for respiratory virus infections reflects their potential therapeutic effects. Rapamycin, an mTOR inhibitor, which is among the most extensively researched autophagy inducers [224], has shown potential in reducing SARS-CoV-2 replication in kidney epithelial cells and acute lung injury cultures [225], as well as reducing MERS-CoV virus replication [116]. In contrast, chloroquine and hydroxychloroquine, known autophagy inhibitors, have been explored for their potential to block the replication of respiratory viruses such as SARS-CoV-2 [159].

However, clinical trials have yielded mixed results, and the safety profile of these drugs in long-term use remains a concern. In recent years, there have been increases in clinical trials investigating autophagy-modulating therapies in the treatment of respiratory viral diseases, particularly during the COVID-19 pandemic.

### 7.1. Current Clinical Trials Targeting Autophagy in Respiratory Virus Infections

Several trials have focused on repurposing existing drugs that modulate autophagy. Notably, azithromycin has been found to interfere with autophagic flux by impairing the lysosomal function [226]. Specifically, it increases the number of autophagosomes while simultaneously blocking their degradation, leading to the accumulation of autophagic vesicles within the cells [226]. The modulation of autophagy by azithromycin may contribute to its effectiveness against certain viral infections. Consequently, a trial is investigating the use of azithromycin as a potential treatment for severe RSV infections in pediatric patients [227].

On the other hand, some studies suggest that antiviral treatments may induce autophagy, thereby contributing to the clearance of viral particles. For example, oseltamivir, an antiviral treatment long used against influenza virus, significantly increased autophagy, as revealed by the significantly higher ratios of LC3-II/LC3-I, increased expression of Beclin1, and decreased expression of p62 [228].

An upcoming trial aims to evaluate the antiviral effects in low-risk patients with high viral loads and uncomplicated influenza infection, with the goal of determining in vivo antiviral activity. In this context, various influenza antivirals, such as oseltamivir, peramivir, zanamivir, laninamivir, baloxavir, and favipiravir will be assessed both individually and in combination. This study seeks to address the current lack of direct comparisons between these antiviral treatments [229].

Moreover, studies have explored the potential of rapamycin to reduce the severity of influenza virus infections [230]. Rapamycin enhances the clearance of influenza viruses from infected cells by inhibiting the mTOR pathway [231]. In another trial, oseltamivir is being evaluated in combination with an autophagy modulator, such as rapamycin, to determine whether this combination enhances the antiviral effects against influenza. Autophagy induction may increase the effectiveness of the drug in reducing viral replication [232]. Rapamycin (Sirolimus), an mTOR inhibitor, modulates autophagy and has shown potential antiviral and anti-fibrotic effects [233]. This is particularly relevant in the context of mitigating the progression to pulmonary fibrosis, a severe complication of COVID-19 [234].

These trials represent diverse approaches to harness autophagy for therapeutic benefit, ranging from repurposing existing drugs to exploring novel dietary interventions.

Additionally, angiotensin II (Ang II) promotes autophagy by upregulating key autophagy-related proteins, including Beclin1, VPS34, Atg12-Atg5, Atg4, and Atg7, and enhances Beclin1 phosphorylation [235]. ACE inhibitors (ACEIs) are known to suppress this pathway, and a clinical trial is currently investigating the therapeutic potential of ACEIs as a treatment for COVID-19, based on their ability to modulate these mechanisms [236].

Table 4 summarizes the ongoing clinical trials targeting autophagy in viral respiratory disorders. Modulation of autophagy has emerged as a promising therapeutic strategy for various diseases, particularly viral infections. As research continues to advance, the intricate interplay between autophagy modulation and disease mechanisms is progressively being unraveled, offering new avenues for therapeutic strategy development.

### 7.2. Future Directions for Therapeutic Strategies

Given the global burden of these viruses, understanding how they manipulate autophagy to enhance their replication and evade immune responses is critical for identifying novel therapeutic strategies. Future therapeutic strategies should aim to fine-tune the modulation of autophagy to maximize the antiviral effects while minimizing potential adverse outcomes. Combination therapies that include autophagy modulators with antiviral drugs may provide a more balanced approach, targeting multiple aspects of viral replication and immune response.

One promising area of research is the development of selective autophagy modulators that specifically target the autophagic machinery used by viruses, without broadly affecting the host autophagic process [21]. This approach could prevent viruses from hijacking the autophagic process, while preserving the beneficial aspects of autophagy for the host [21,237].

Further research in this area could be highly impactful and contribute to a better understanding of the interactions between viruses and autophagy.

## 8. Conclusions

In conclusion, the extensive review of autophagy’s role in the interaction between respiratory viruses and their host cells has revealed a complex landscape of cellular mechanisms, where autophagy functions as both an accomplice to viruses and a potent defense strategy of the host. This duality not only underscores the sophisticated nature of the viral–host interactions but also illuminates significant opportunities for therapeutic interventions. By exploiting the dual roles of autophagy (hindering viral exploitation of cellular processes and bolstering the antiviral immune response), we can develop innovative treatments that are both effective and precise.

The potential for modulating autophagy in the treatment of respiratory viral infections offers a promising frontier in drug development. Focused research into the specific molecular mechanisms by which various respiratory viruses manipulate autophagy could lead to the creation of targeted therapies that selectively inhibit these viral strategies without compromising the host’s vital defenses. The development of specific autophagy modulators capable of distinguishing between viral components and host cell proteins could pave the way for treatments that minimize side effects and circumvent the issue of viral resistance.

Moreover, integrating autophagy modulators with established antiviral agents could significantly enhance therapeutic outcomes, reducing viral load and tempering the inflammatory responses that are hallmarks of severe viral infections. This combination therapy approach could decrease both morbidity and mortality among affected populations, presenting a robust response to the global health challenges posed by respiratory viruses.

As we look to the future, the intersection of molecular research and pharmaceutical innovation holds great promise for pandemic preparedness. By fostering collaborative efforts that bridge academic research and clinical applications and leveraging cutting-edge technologies like artificial intelligence in drug discovery, we can accelerate the development of effective therapies. These endeavors will not only address current health crises but also strengthen global capacities to manage future pandemics, ensuring swift, scalable, and equitable health responses across diverse populations.

By embracing these strategies, we commit to a future where advanced understanding and innovative application of autophagy in viral therapy opens new avenues for combating respiratory viruses, thus safeguarding global health and advancing medical science.

## Figures and Tables

**Figure 1 cells-14-00418-f001:**
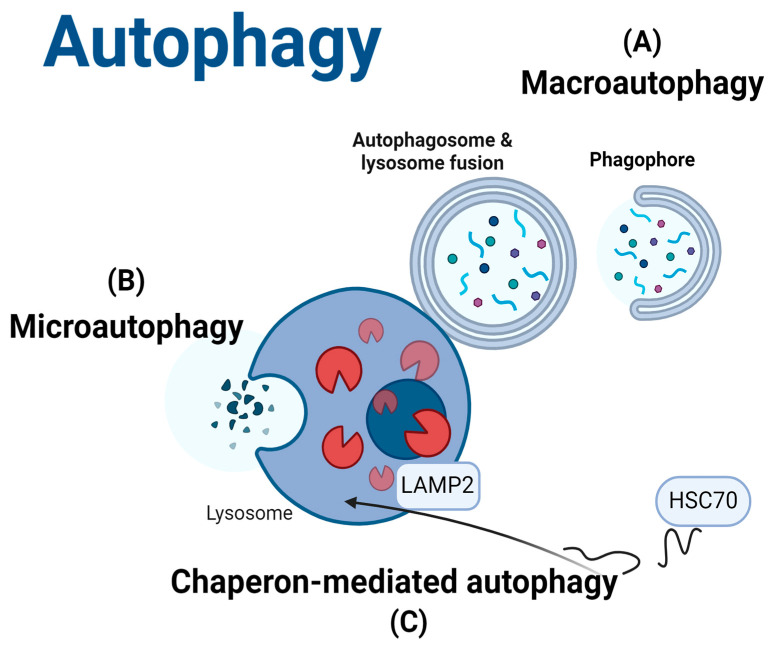
Schematic illustration of autophagy pathways. (**A**) Macroautophagy (autophagy), which includes phagophore formation and expansion, autophagosome and lysosome fusion, and cargo degradation; (**B**) microautophagy, the lysosome takes up soluble particulates by protrusion or invagination; and (**C**) CMA, a selective degradation mechanism for specific proteins. This figure was created with BioRender.com. Licensing Right: VB27W582O8.

**Figure 2 cells-14-00418-f002:**
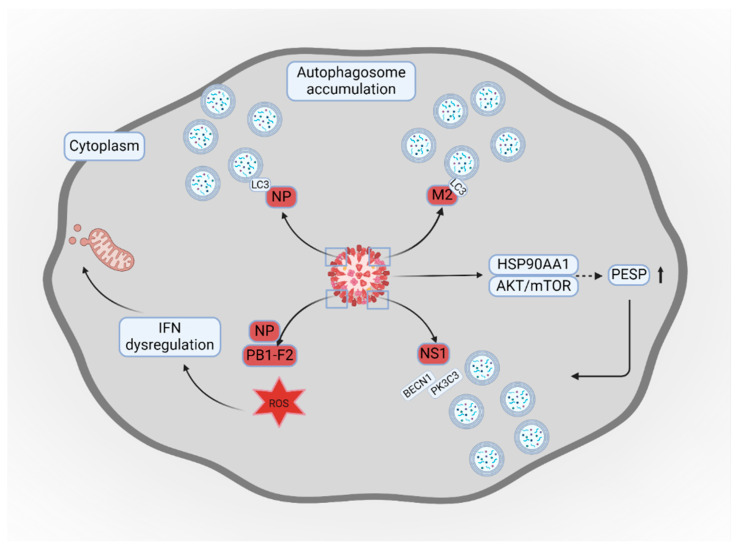
Schematic illustration of influenza virus interaction with autophagy. Influenza virus induces mitochondrial damage through viral NP and PB1-F2 proteins by release of ROS and IFN dysregulation that induces viral replication. The NP and M2 viral proteins, by attachment to LC3 and NS1 proteins through the interaction of BECN1 with PK3C3, induce autophagosome accumulation. Upregulation of HSP90AA1 and the involvement of the AKT/mTOR signaling pathway following viral infection cause autophagosome accumulation. This figure was created with BioRender.com. Licensing Right: HP27RKPZNV.

**Figure 3 cells-14-00418-f003:**
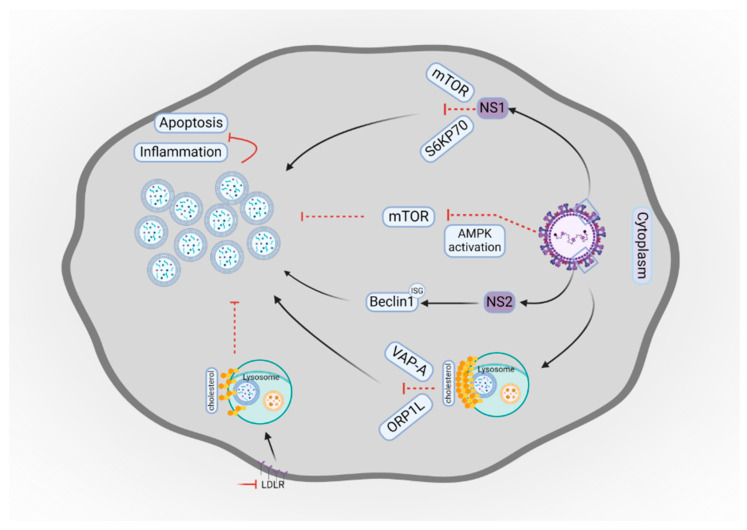
Schematic illustration of RSV interaction with autophagy. The RSV NS1 protein inhibits the mTOR-S6KP70 signaling pathway that triggers autophagosome accumulation. The NS2 protein stabilizes Beclin1 via ISGylation. This interaction causes the successful induction of autophagosome accumulation. RSV may cause cholesterol accumulation in lysosomes and weaken VAP-A and ORP1L binding which enables autophagosome accumulation. AMPK activation during RSV infection induces autophagy by inhibiting the mTOR pathway and activation of autophagosome degradation. The LDLR knock out inhibits RSV infection by mediating lysosomal cholesterol metabolism and autophagy. The production of inflammatory cytokines and the activation of apoptosis are inhibited by the enhanced process of autophagy. This figure was created with BioRender.com. Licensing Right: KV27RKQMGG.

**Figure 4 cells-14-00418-f004:**
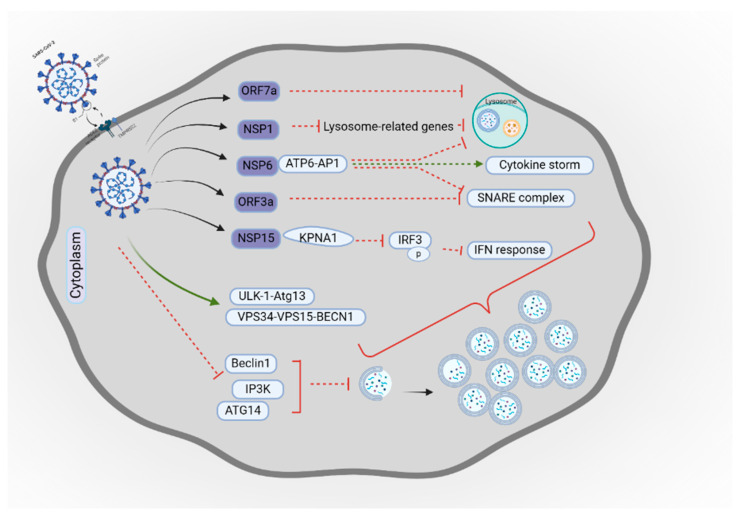
Schematic illustration of SARS-CoV-2 interaction with autophagy. SARS-CoV-2 starts its life cycle via interaction with ACE2. The virus activates ULK-1-Atg13 and VPS34-VPS15-BECN1 complexes that cause autophagosome accumulation, downregulates BECN1, IP3K, and ATG14, and inhibits phagophore formation. The ORF7a decreases lysosomal acidity. Nsp1 downregulates the lysosome-related genes and decreases lysosomal acidity. NSP6 disrupts lysosome acidification through interaction with ATP6AP1. NSP6, by interaction with ATP6AP1, blocks the assembly of the SNARE complex and triggers a cytokine storm (ARDS). Nsp15 promotes the degradation of KPNA1 and inhibits type I interferon response by inhibiting phosphorylated IRF3. ORF3a blocks the assembly of the SNARE complex. All these events lead to autophagosome accumulation. This figure was created with BioRender.com. Licensing Right: GN27RKQUQC.

**Figure 5 cells-14-00418-f005:**
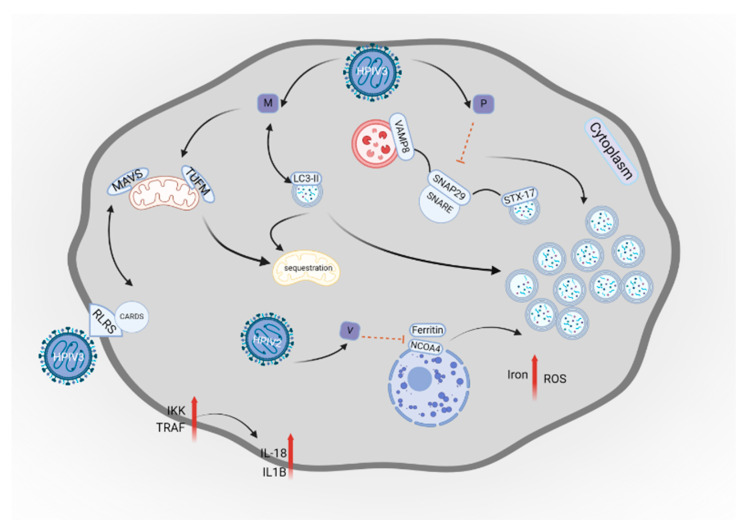
Schematic illustration of HPIV interaction with autophagy. HPIV3 P protein blocks degradation of autophagosomes. STX17 interacts with SNAP29 protein in the SNARE complex. SNAP29 interacts with VAMP8 in the lysosome membrane. P protein binds to SNAP29 and prevents its interaction with STX17, thus inhibiting autophagosome degradation. HPIV3 is recognized by RLRs with CARD, which binds with MAVS and recruits IKK and TRAF, which in turn causes production of IFN and pro-inflammatory cytokines. The M protein of HPIV3 binds to TUFM and causes mitochondrial sequestration. It mediates the formation of autophagosomes by interaction with LC3. HPIV2 V protein suppresses the interaction between NCOA4 and ferritin, allowing the virus to grow effectively in an environment rich in iron and ROS. Insufficient autophagy leads to the accumulation of autophagosomes, which elevates extracellular viral production. This figure was created with BioRender.com. Licensing Right: LT27RKR4W1.

**Figure 6 cells-14-00418-f006:**
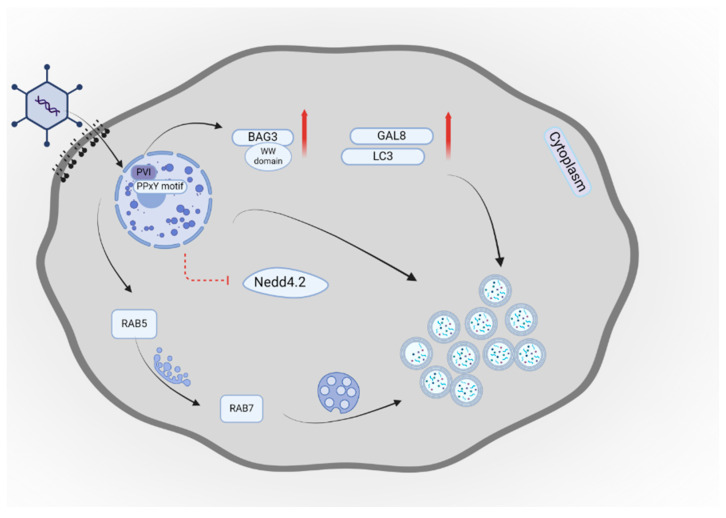
Schematic illustration of AdV interaction with autophagy. The PVI of the AdV damages the membrane and allows the AdV to enter the cytosol and nucleus. Gal-8 and LC3 proteins recruit impaired endosomes, which induce an autophagic response. The host BAG3 WW domain interacts with PVI. Additionally, PVI promotes the expression of BAG3. The AdV escapes autophagy with Rab5–Rab7 exchange during the transition from early to late endosomes. The PPxY motif of PVI sequestrates Nedd4.2, a ubiquitin ligase preventing autophagosome development and enhancing the infectivity. This figure was created with BioRender.com. Licensing Right: QO27RKRE2Z.

**Table 1 cells-14-00418-t001:** The most common respiratory viruses and their autophagy-related mechanisms.

Virus	Family	Mechanisms
Influenza A virus (IAV)	Orthomyxoviridae	M2 ion channel blocks autophagosome–lysosome fusion [19,20]
SARS-CoV-2	Coronaviridae	ORF3a blocks autophagosome–lysosome fusion; NSP6 limits autophagosome expansion [21,22]
Respiratory Syncytial Virus (RSV)	Pneumoviridae	NS2 protein activates autophagy through BECN1 [23,24]
Parainfluenza Virus (PIV)	Paramyxoviridae	Phosphoprotein P induces incomplete autophagy, blocking autophagosome–lysosome fusion [25,26]
Adenovirus	Adenoviridae	E1B-19K protein interacts with BECN1 to regulate autophagy [27,28]

**Table 2 cells-14-00418-t002:** A summary of recently investigated agents that modulate autophagy and influenza.

Agent	Impact on influenza infection	Impact on autophagy	References
Gallic acid	Decreases viral load	Reduces accumulation of autophagosomes	[86]
Vitamin D3	Induces cytoprotective effects	Enhances fusion of autophagosome and lysosome, thus decreasing viral replication	[87]
Baicalin	Improves viability of infected macrophages	Reduces expression of autophagy marker	[88]
Tanreqing	Inhibits influenza replication	Enhances fusion of autophagosome and lysosome	[65]
Huanglian-Ganjiang combination	Suppresses inflammatory responses	Enhances fusion of autophagosome and lysosome	[89]

**Table 3 cells-14-00418-t003:** Summary of the therapeutic medications targeting autophagy in coronaviruses.

Category	Medications	Mechanism	References
Lysosomotropic Agents	Chloroquine, Hydroxychloroquine	Increases the pH within lysosomes/blocks entry mechanisms of virus/does not inhibit infection of human lung cells with SARS-CoV-2; also blocks some viruses’ biosynthetic processes after entry	[161,166,167]
	Azithromycin	Synergistic effect of hydroxychloroquine and azithromycin on the reduction of viral load of SARS-CoV-2	[168]
	Artemisinin	Targets the Lys353 and Lys31 binding hotspots on the viral spike protein/NF-κB inhibition/blocks SARS-CoV-2 infection	[169,170,171]
	Imatinib	Inhibits fusion of the virions at the endosomal membrane	[172]
Protease Inhibitors	Lopinavir/Ritonavir	Inhibits viral protease/reduction in viral load	[163,164]
	Teicoplanin	Suppresses the proteolytic activity of cathepsin L on Spike/prevents the entry of SARS-CoV-2 into the cytoplasm	[173,174]
PI3K/mTOR Regulators	Rapamycin	Inhibits mTORC1/inhibits protein synthesis/reduces viral replication/reduces MERS-CoV and SARS-CoV-2 infection by activating autophagy	[115]
	Everolimus	Induces autophagy by blocking mTORC1/inhibits MERS-CoV infection	[165]
	Nitazoxanide	Stimulates autophagy by blocking mTORC1/inhibits replication of MERS-CoV and SARS-CoV-2	[175,176]
	Wortmannin	Suppresses autophagy by inhibition of PI3K/inhibits MERS-CoV infection	[165]

**Table 4 cells-14-00418-t004:** Summary of the ongoing clinical trials targeting autophagy in viral respiratory disorders.

Trial Identifier	Activity	Intervention	Phase	Primary Outcome	Link
NCT05060705	COVID-19	Efesovir in comparison with the drug Remdesivir	Phase 2	Reduction of viral load in COVID-19 patients	https://clinicaltrials.gov/study/NCT05060705 (Accession date: 10 December 2024)
NCT05218356	COVID-19	Codivir	Phase 2	Efficacy in reducing the severity of COVID-19	https://clinicaltrials.gov/study/NCT05218356 (Accession date: 10 December 2024)
NCT06128967	COVID-19	Metformin/Fluvoxamine	Phase 3	Evaluation of treatment efficacy in long COVID patients	https://clinicaltrials.gov/study/NCT06128967 (Accession date: 10 December 2024)
NCT06147050	COVID-19	Metformin	Phase 3	Assessment of Chronic Fatigue Syndrome in long COVID patients	https://clinicaltrials.gov/study/NCT06147050 (Accession date: 10 December 2024)
NCT04345406	COVID-19	ACE inhibitors	Phase 3	Clinical efficacy in COVID-19 treatment	https://clinicaltrials.gov/study/NCT04345406 (Accession date: 10 December 2024)
NCT04948203	COVID-19	Sirolimus	Phase2 Phase 3	Prevention of post-COVID-19 fibrosis in hospitalized patients	https://clinicaltrials.gov/study/NCT04948203 (Accession date: 10 December 2024)
NCT06024096	Influenza	Atorvastatin	Phase 4	Effect of statins on influenza vaccine response	https://clinicaltrials.gov/study/NCT06024096 (Accession date: 10 December 2024)
NCT05026749	RSV	Azithromycin	Phase 3	Efficacy in RSV-induced respiratory failure in children	https://clinicaltrials.gov/study/NCT05026749 (Accession date: 10 December 2024)
NCT03901001	Influenza	Sirolimus + Oseltamivir vs. Oseltamivir Alone	Phase 3	Comparison of treatment outcomes for influenza	https://clinicaltrials.gov/study/NCT03901001 (Accession date: 10 December 2024)
NCT05648448	Influenza	Influenza antivirals	Phase 2	Assessing antiviral efficacy in early symptomatic influenza	https://clinicaltrials.gov/study/NCT05648448 (Accession date: 10 December 2024)

## Data Availability

No new data were created or analyzed in this study.

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
