# Peer review of "Autophagy and Respiratory Viruses: Mechanisms, Viral Exploitation, and Therapeutic Insights"

_cells, 2025, doi:10.3390/cells14060418_

Round 1
Reviewer 1 Report
Comments and Suggestions for Authors
This manuscript provides a comprehensive review of the interplay between autophagy and respiratory viral infections. The authors summarize the current understanding of how autophagy is modulated by various respiratory viruses and, in turn, influences the course of infection. The review covers a broad range of respiratory viruses, including influenza virus, respiratory syncytial virus (RSV), coronaviruses, human parainfluenza viruses (HPIVs), and adenoviruses. It concludes with a discussion of potential therapeutic strategies targeting autophagy.
However, as I am not an expert in autophagy, my comments will primarily focus on the aspects related to RSV.
1. The discussion on the potential role of traditional Chinese medicine, Qingfei Oral Liquid, in alleviating RSV-associated inflammation is intriguing and offers a novel perspective. However, since the pharmacological mechanisms of traditional Chinese medicine are not yet well-defined, could the authors explore whether there are chemical drugs with more clearly defined pharmacology and targets that could modulate RSV-associated autophagy pathways to mitigate inflammation?
2. It is now well-established that IGF1R serves as a primary receptor for RSV and that IGF1R is involved in regulating immune pathways through various mechanisms. Could the authors discuss the potential impact of RSV-induced activation of the IGF1R pathway? This may help to provide insights into some recent challenges associated with preF antigen vaccines.
This manuscript provides a valuable contribution to the field of respiratory virology by highlighting the role of autophagy in viral infection. The suggested revisions, particularly regarding the critical assessment of traditional medicine and the dedicated discussion of the IGF1R pathway in RSV infection, will further strengthen the manuscript.
Author Response
- The discussion on the potential role of traditional Chinese medicine, Qingfei Oral Liquid, in alleviating RSV-associated inflammation is intriguing and offers a novel perspective. However, since the pharmacological mechanisms of traditional Chinese medicine are not yet well-defined, could the authors explore whether there are chemical drugs with more clearly defined pharmacology and targets that could modulate RSV-associated autophagy pathways to mitigate inflammation?
- Answer: We sincerely appreciate the Reviewer’s insightful suggestion regarding the potential use of chemical drugs with well-defined pharmacological mechanisms to modulate RSV-associated autophagy and mitigate inflammation. In addition to Qingfei (QF) oral liquid, which has demonstrated inhibitory effects on autophagosome formation and RSV-associated inflammation, several pharmacological agents have been investigated for their ability to regulate autophagy in viral infections and may hold promise for RSV treatment.
- Among them, rapamycin, a well-characterized mTORC1 inhibitor and autophagy activator, has been shown to suppress excessive inflammatory responses in viral infections, including influenza and SARS-CoV-2. Studies have demonstrated that rapamycin treatment reduces pulmonary inflammation, cytokine storms, and immune cell infiltration, supporting its potential as a modulator of virus-induced autophagy-related inflammation. However, its effect on RSV infection remains largely unexplored, and existing studies suggest that rapamycin may enhance viral replication and impair dendritic cell-mediated immune responses, indicating that its therapeutic use in RSV would require careful evaluation.
- Another promising agent is chloroquine (CQ), a known autophagy inhibitor that prevents autophagosome-lysosome fusion. It has demonstrated antiviral effects by increasing endosomal pH and preventing viral entry, as reported in SARS-CoV-2 studies. While CQ effectively disrupts viral replication in vitro, its applicability to RSV remains unclear, necessitating further investigation into whether CQ-mediated autophagy blockade could modulate RSV-induced inflammation without adverse effects on viral clearance.
- Additionally, metformin, an AMPK activator, has garnered attention for its ability to regulate mTOR signaling, autophagy, and viral infection dynamics. In the context of SARS-CoV-2, metformin has been shown to reduce viral entry by modulating ACE2 receptor phosphorylation and endosomal pH regulation. Its ability to modulate autophagic flux and inflammatory cytokine production suggests that it could have a potential role in RSV-associated inflammation. However, studies evaluating its impact on RSV are currently lacking, and further research is required to determine its efficacy and mechanism of action in this specific context.
- In light of these findings, we acknowledge that several drugs with well-characterized autophagy-modulating properties exist, but their role in RSV infection remains largely unexplored. We have incorporated relevant discussion and references into the manuscript to highlight these chemical agents as potential alternatives to traditional Chinese medicine, emphasizing the need for further studies to assess their efficacy in RSV-associated autophagy and inflammation (lines 374–417).
- Once again, we appreciate the Reviewer's valuable input, which has strengthened our discussion on pharmacological interventions targeting RSV-associated autophagy pathways.
- It is now well-established that IGF1R serves as a primary receptor for RSV and that IGF1R is involved in regulating immune pathways through various mechanisms. Could the authors discuss the potential impact of RSV-induced activation of the IGF1R pathway? This may help to provide insights into some recent challenges associated with preF antigen vaccines.
- Answer: We sincerely appreciate the Reviewer’s suggestion regarding the role of IGF1R activation in RSV infection and its potential implications for preF antigen vaccines. The insulin-like growth factor-1 receptor (IGF1R) has been identified as a primary receptor for RSV, playing a crucial role in viral entry and immune regulation. Current evidence suggests that RSV binding to IGF1R stimulates nucleolin (NCL) trafficking to the cell membrane, facilitating viral attachment and entry. This process is mediated by protein kinase C zeta (PKCζ), which enhances viral uptake. Interestingly, despite its role in viral entry, RSV downregulates IGF1R expression in human airway epithelial cells during infection, a phenomenon thought to facilitate viral release and further infection spread.
- Beyond its role in viral entry, IGF1R activation triggers downstream signaling pathways, including phosphoinositide 3-kinase (PI3K), which has been implicated in actin remodeling and macropinocytosis-mediated RSV uptake. The involvement of IGF1R in RSV pathogenesis presents a potential therapeutic target, as studies have demonstrated that inhibiting IGF1R suppresses viral entry. For instance, CL-A3-7, a small molecule inhibitor, effectively blocks RSV infection by downregulating IGF1R and viral F protein expression, thereby limiting viral replication. These findings highlight IGF1R as a key regulator of RSV-host interactions and a potential antiviral target.
- Given the importance of the RSV F protein in vaccine development, particularly in preF antigen-based vaccines, the modulation of IGF1R signaling may have implications for vaccine efficacy. The preF antigen vaccine works by stimulating the production of antibodies that recognize the prefusion conformation of the RSV F protein, enhancing immune responses against viral entry. Since IGF1R inhibition has been shown to suppress F protein expression, it is conceivable that combining preF vaccination with IGF1R-targeting therapies could enhance antiviral responses by simultaneously preventing viral entry and promoting immune clearance. Additionally, Ziresovir, a selective RSV F protein inhibitor, effectively blocks viral penetration into human cells and has demonstrated a favorable safety and efficacy profile in clinical trials. Whether F protein inhibitors could be combined with IGF1R inhibition remains an open question, but such a dual-targeting approach may provide a novel strategy for RSV prevention and treatment.
- In response to the Reviewer's comment, we have incorporated a discussion on the impact of RSV-induced IGF1R activation and its potential interplay with preF antigen vaccines, along with relevant references, into the revised manuscript (lines 321–351). We appreciate this valuable suggestion, as it strengthens the discussion on RSV entry mechanisms and potential therapeutic interventions.
Reviewer 2 Report
Comments and Suggestions for Authors
The article by Aligolighasemabadi et al. explores the dual role of autophagy in respiratory viral infections, highlighting how viruses exploit and manipulate autophagic pathways while identifying therapeutic potential in targeting these interactions.
- Based on the data review in the article, the best title will be "Autophagy and Respiratory Viruses: Mechanisms, Viral Exploitation, and Therapeutic Insights."
- In the introduction, more specific data about the relevance of each virus must be added.
- Avoid using" the elderly." Change by older adults.
- Figure 1 must be improved since macroautophagy is not visible according to the legend. It would be more explicative if microautophagy were separately explained from macroautophagy (Fig. A and B).
- Table 1. Respiratory syncytial virus. Parainfluenza virus. Also, include more references for each virus.
- Lines 119-124. Poorly described and cited.
- The function of LC3 must be explained.
- Explain the role of LRPPRC, BECN1, and PK3C3.
- Why H5N1 is a variant and H9N2 is a strain?. The author must explain.
- The paragraph in lines 153-169 must be improved since it is not well explained and connected.
- Line 171. What avian influenza?
- The therapeutic approach must be improved and correlate with table 2.
- Line 224: RSV does not belong the the paramyxoviridae family.
- The RSV section must be improved since there is enough data regarding therapeutic and the role in inhibing autophagy.
- The coronavirus section must be improved ar describe in detail the therapeutic agents described in table 3.
-
Comments on the Quality of English Language
The English quality is acceptable, but grammar must be improved.
Author Response
- Based on the data review in the article, the best title will be "Autophagy and Respiratory Viruses: Mechanisms, Viral Exploitation, and Therapeutic Insights."
- Answer: Thanks for the important suggestion. The title was updated as suggested.
- In the introduction, more specific data about the relevance of each virus must be added.
- Answer: We appreciate the Reviewer's suggestion to provide more specific data regarding the relevance of each virus in the introduction. In response, we have added quantitative data on the global burden of influenza, RSV, COVID-19, parainfluenza, and adenoviruses to highlight their significance in respiratory infections.
- Specifically, we now include that influenza affects 3–5 million individuals with severe illness annually, resulting in 290,000–650,000 deaths worldwide. RSV is identified as the leading cause of acute lower respiratory tract infections in children under five and older adults, responsible for over 33 million cases and approximately 120,000 deaths globally each year, imposing a substantial healthcare burden. Additionally, we have incorporated data on COVID-19’s extensive impact, resulting in millions of infections and hundreds of thousands of deaths worldwide. Furthermore, we have highlighted the roles of parainfluenza viruses, which pose a major risk to immunocompromised individuals and older adults, and adenoviruses, which can cause severe respiratory distress, particularly in military populations and young children.
- Avoid using" the elderly." Change by older adults.
- Answer: All “the elderly” phrases were replaced with “older adults” throughout the manuscript.
- Figure 1 must be improved since macroautophagy is not visible according to the legend. It would be more explicative if microautophagy were separately explained from macroautophagy (Fig. A and B).
- Answer: Figure 1 and its legend were updated and clearly distinguishes macroautophagy from microautophagy and CMA.
- Table 1. Respiratory syncytial virus. Parainfluenza virus. Also, include more references for each virus.
- Answer: Table 1 was reviewed and improved by addition of more references.
- Lines 119-124. Poorly described and cited.
- Answer: The information was updated and cited relatedly.
- The function of LC3 must be explained.
- Answer: The function of LC3 was explained comprehensively.
- Explain the role of LRPPRC, BECN1, and PK3C3.
- Answer: These key proteins explanations were expanded in the text.
- Why H5N1 is a variant and H9N2 is a strain?. The author must explain.
- Answer: We sincerely appreciate the Reviewer’s insightful comment regarding the distinction between H5N1 as a variant and H9N2 as a strain. This distinction is crucial in virology, as the terms "variant" and "strain" are often used interchangeably but have specific definitions based on genetic and phenotypic characteristics.
- A variant refers to a virus that has undergone genetic mutations compared to its original reference genome, but these mutations do not necessarily result in significant functional or behavioral changes. In contrast, a strain is a variant that has accumulated genetic changes leading to distinct phenotypic characteristics, such as differences in transmissibility, virulence, host range, or antigenicity. While all strains are variants, not all variants qualify as strains unless they exhibit functional divergence.
- In the context of avian influenza viruses (AIVs), H5N1 is considered a variant because it represents a genetically mutated form of earlier H5 lineage viruses, with modifications that influence its transmissibility and pathogenicity. However, it retains the fundamental biological properties of the H5 hemagglutinin (HA) subtype, making it a variant rather than an entirely distinct strain.
- On the other hand, H9N2 is classified as a strain because it has established a unique epidemiological and pathogenic profile distinct from other H9 subtype viruses. H9N2 has been shown to circulate widely in poultry populations, occasionally infecting humans, and has reassorted with other influenza viruses, contributing to the evolution of new zoonotic threats. Its distinct biological behavior and sustained transmission in avian and mammalian hosts justify its classification as a strain rather than just a variant.
- To ensure clarity and scientific accuracy, we have maintained the use of "variant" for H5N1 and "strain" for H9N2, aligning with standard virological definitions
- https://www.health.qld.gov.au/newsroom/features/what-are-virus-mutations-variants-and-strains
Therefore we kept “variant” instead of “strain”.
- The paragraph in lines 153-169 must be improved since it is not well explained and connected.
- Answer: The sentences were rephrased to explain more clearly the autophagy modulation on immune responses across different viruses and link it to therapeutic applications.
- Line 171. What avian influenza?
- Answer: “avian influenza H7N9 subtype” was added.
- The therapeutic approach must be improved and correlate with Table 2.
- Answer: The therapeutic approach was revised, improved, and correlated with Table 2.
- Line 224: RSV does not belong the Paramyxoviridae family.
- Answer: The typo mistake was corrected.
- The RSV section must be improved since there is enough data regarding therapeutic and the role in inhibiting autophagy.
- Answer: The RSV section was comprehensively improved.
- The coronavirus section must be improved and describe in detail the therapeutic agents described in table 3.
- Answer: The coronavirus section was improved.
The English quality is acceptable, but grammar must be improved.
- Answer: Thanks for the improvement suggestion. Grammar and language check has been conducted, ensuring clarity and accuracy throughout the manuscript.
Reviewer 3 Report
Comments and Suggestions for Authors
Summary
The authors point out that many respiratory viruses continue to impose a substantial public health burden, in spite of extant vaccination programmes. Therapeutic options, however, are relatively limited, suggesting a need for novel treatment strategies. They suggest that autophagy, specifically macroautophagy, has emerged as a host response cellular process to respiratory infections that, because it degrades damaged organelles and pathogens and enables xenophagy with selective targeting of viral particles, enhances cellular defence. Viruses have evolved evasive mechanisms that involve the manipulation of autophagy, thereby permitting viral replication. Their current paper provides a review of the dual role of autophagy in viral manipulation and host response, and the complex interplay between respiratory viruses and autophagy - related pathways. Their aim is to highlight the therapeutic potential of targeting autophagy to enhance antiviral responses, paving the way to effective treatments to viral infections.
General comments
The introduction is well-written and appropriately referenced with recent publications, and provides a clear and concise rationale for the current review. One suggestion, however, is to provide a concise definition of autophagy and macroautophagy earlier in the introduction, which might retain/expand a broader audience for this substantive review. The one provided, including Figure 1, from lines 90 through 102 is comprehensive and easy to introduce readers to the topic.
The authors provide a detailed, virus-specific review of the known mechanisms by which viruses exploit autophagy in order to suppress or otherwise evade host immune defence. These are detailed and well-referenced with recent studies, well-organised with summaries at the end of each pathogen-specific section. Areas requiring further elucidation are identified. For example, with influenza they highlight the dual role of autophagy, both as a means of viral suppression of host response and host cell response to counteract the virus, as well as identifying specific pharmacological agents that can modulate specific autophagy pathways for host defence.
Of note are the references to traditional medicines, which contain some of these modulating agents, in addition to vitamins, like D3 (apparently a fusion enhancer of autophagosmes and lysosomes against influenza replication).
The figures provide well thought out illustrations of these effects, and aid in making the mechanisms of action and targets for intervention clearer, and provide effective complements to the tables.
If there is any concern about the manuscript it is that is dense reading, however many will focus on specific pathogens and/or mechanisms of action for developing efficacious interventions. Having this information in one location will facilitate interdisciplinary and complementary collaboration, where possible, against more than one virus.
The above lays the groundwork for the introduction of ongoing clinical trials that the authors tabulate under ‘Potential Therapeutic Approaches’, listing several current clinical trials. Their foregoing treatment of the subject matter provides a clear and rational basis for these trials, and sets the stage for a potential follow up review of the various trial outcomes and potential future therapeutic directions.
Specific comments
- This reviewer was unable to retrieve citation 14 (Zhou et al. 2022)
Author Response
The introduction is well-written and appropriately referenced with recent publications, and provides a clear and concise rationale for the current review. One suggestion, however, is to provide a concise definition of autophagy and macroautophagy earlier in the introduction, which might retain/expand a broader audience for this substantive review. The one provided, including Figure 1, from lines 90 through 102 is comprehensive and easy to introduce readers to the topic.
- Answer: Thanks for the suggestion. The definition of autophagy and macroautophagy, already present in lines 91–130, has been highlighted for clarity and better visibility.
The authors provide a detailed, virus-specific review of the known mechanisms by which viruses exploit autophagy in order to suppress or otherwise evade host immune defense. These are detailed and well-referenced with recent studies, well-organized with summaries at the end of each pathogen specific section. Areas requiring further elucidation are identified. For example, with influenza they highlight the dual role of autophagy, both as a means of viral suppression of host response and host cell response to counteract the virus, as well as identifying specific pharmacological agents that can modulate specific autophagy pathways for host defense.
- Answer: The virus-specific sections have been expanded to clarify autophagy dual role in both viral exploitation and host defense. Pharmacological agents targeting autophagy have been thoroughly discussed.
Of note are the references to traditional medicines, which contain some of these modulating agents, in addition to vitamins, like D3 (apparently a fusion enhancer of autophagosmes and lysosomes against influenza replication).
- Answer: The discussion on traditional medicine and vitamin D3 has been expanded with additional references to highlight their roles in autophagy modulation.
The figures provide well thought out illustrations of these effects, and aid in making the mechanisms of action and targets for intervention clearer, and provide effective complements to the tables.
- Answer: All Figures were double-checked and updated if needed. All Tables were reviewed and improved for accuracy.
If there is any concern about the manuscript it is that is dense reading, however many will focus on specific pathogens and/or mechanisms of action for developing efficacious interventions. Having this information in one location will facilitate interdisciplinary and complementary collaboration, where possible, against more than one virus.
The above lays the groundwork for the introduction of ongoing clinical trials that the authors tabulate under ‘Potential Therapeutic Approaches’, listing several current clinical trials. Their foregoing treatment of the subject matter provides a clear and rational basis for these trials, and sets the stage for a potential follow up review of the various trial outcomes and potential future therapeutic directions.
- Answer: The clinical trials and therapeutic approaches have been verified for accuracy, and corresponding references to therapeutic agents have been added in the main text.
Specific comments
- This reviewer was unable to retrieve citation 14 (Zhou et al. 2022).
- Answer: Kindly, the reference was checked and it was retrievable:
(https://pmc.ncbi.nlm.nih.gov/articles/PMC9794007/pdf/KVIR_13_2014680.pdf)
Zhou A., Zhang W., Dong X., Liu M., Chen H., Tang B. (2022) The battle for autophagy between host and influenza A virus, Virulence. 13(1):46-59